# The leaderless communication peptide (LCP) class of quorum-sensing peptides is broadly distributed among Firmicutes

Shifu Aggarwal[1,2,7], Elaine Huang[1,2,7], Hackwon Do[1,2,3,7], Nishanth Makthal[1,2], Yanyan Li [4], Eric Bapteste[5], Philippe Lopez [5], Charles Bernard [6] ✉ & Muthiah Kumaraswami [1,2] ✉

The human pathogen *Streptococcus pyogenes* secretes a short peptide (leaderless communication peptide, LCP) that mediates intercellular communication and controls bacterial virulence through interaction with its receptor, RopB. Here, we show that LCP and RopB homologues are present in other Firmicutes. We experimentally validate that LCPs with distinct peptide communication codes act as bacterial intercellular signals and regulate gene expression in *Streptococcus salivarius*, *Streptococcus porcinus*, *Enterococcus malodoratus* and *Limosilactobacillus reuteri*. Our results indicate that LCPs are more widespread than previously thought, and their characterization may uncover new signaling mechanisms and roles in coordinating diverse bacterial traits.

Bacteria use quorum sensing (qs) pathways to monitor the population density of their own and closely related competing species and coordinate population-wide phenotypic alterations that aid their survival[1–6]. The qs pathways control critical phenotypic traits such as bacterial virulence, biofilm formation, and antibiotic resistance[1–6]. The qs signaling involves signal production, signal export, signal recognition by extracellular or intracellular receptors, and the subsequent modulation of target gene expression[1,3]. The gram-positive bacteria typically use secreted cyclic or linear oligopeptides to carry out intercellular signaling[6,7]. The Rap-Rgg-NprR-PrgX-PlcR-AimR (RRNPPA)-family regulators in firmicutes and phages constitute the largest family of cytosolic qs receptors that bind to re-internalized linear oligopeptides to mediate gene regulation[7]. Invariably, the peptides are produced as long, inactive propeptides that contain the secretion signal sequence for their export and cleavage sites for membrane-bound or secreted proteases (Fig. 1a). The processed 5–8 amino acid peptides constitute the active intercellular signal that controls gene expression[8,9].

Recently, we discovered a new class of **l**eaderless **c**ommunication **p**eptide (LCP) in human pathogen *Streptococcus pyogenes* that controls bacterial virulence (Fig. 1a)[10]. Contrary to characterized bacterial peptide signals, the LCP, known as SpeB-inducing peptide (SIP), is synthesized from a 24 base pair ultrasmall open reading frame (usORF) encoding a mature 8 amino acid peptide without the amino acid sequences required for secretion and processing[10]. Consistent with the lack of hallmarks of bacterial peptide signals, the canonical machineries required for the export and import of qs peptides are not involved in SIP transport[10]. However, SIP is produced, secreted, and reimported into the bacterial cytosol. The cytosolic SIP binds to a RRNPPA-family transcription activator, Regulator of Protease B (RopB), which leads to the upregulation of an adjacently located virulence factor, a secreted cysteine protease SpeB, and increased bacterial virulence in animal models of infection[10,11].

The lack of conformity of LCPs with the hallmarks of classical bacterial peptide signals and their presence as a 24-base pair usORF

[1]Center for Molecular and Translational Human Infectious Diseases Research, Houston Methodist Research Institute, Houston, TX 77030, USA. [2]Department of Pathology and Genomic Medicine, Houston Methodist Hospital, Houston, TX 77030, USA. [3]Research Unit of Cryogenic Novel Material, Korea Polar Research Institute, Incheon 21990, South Korea. [4]Communication Molecules and Adaptation of Microorganisms (MCAM), CNRS, Muséum National d'Histoire Naturelle, Paris, France. [5]Institut de Systématique, Evolution et Biodiversité (ISYEB), Sorbonne Université, CNRS, Muséum National d'Histoire Naturelle, Paris, France. [6]Department of Computational Biology, University of Lausanne, Lausanne, Switzerland. [7]These authors contributed equally: Shifu Aggarwal, Elaine Huang, Hackwon Do. ✉e-mail: charles.bernard@cri-paris.org; mkumaraswami@houstonmethodist.org

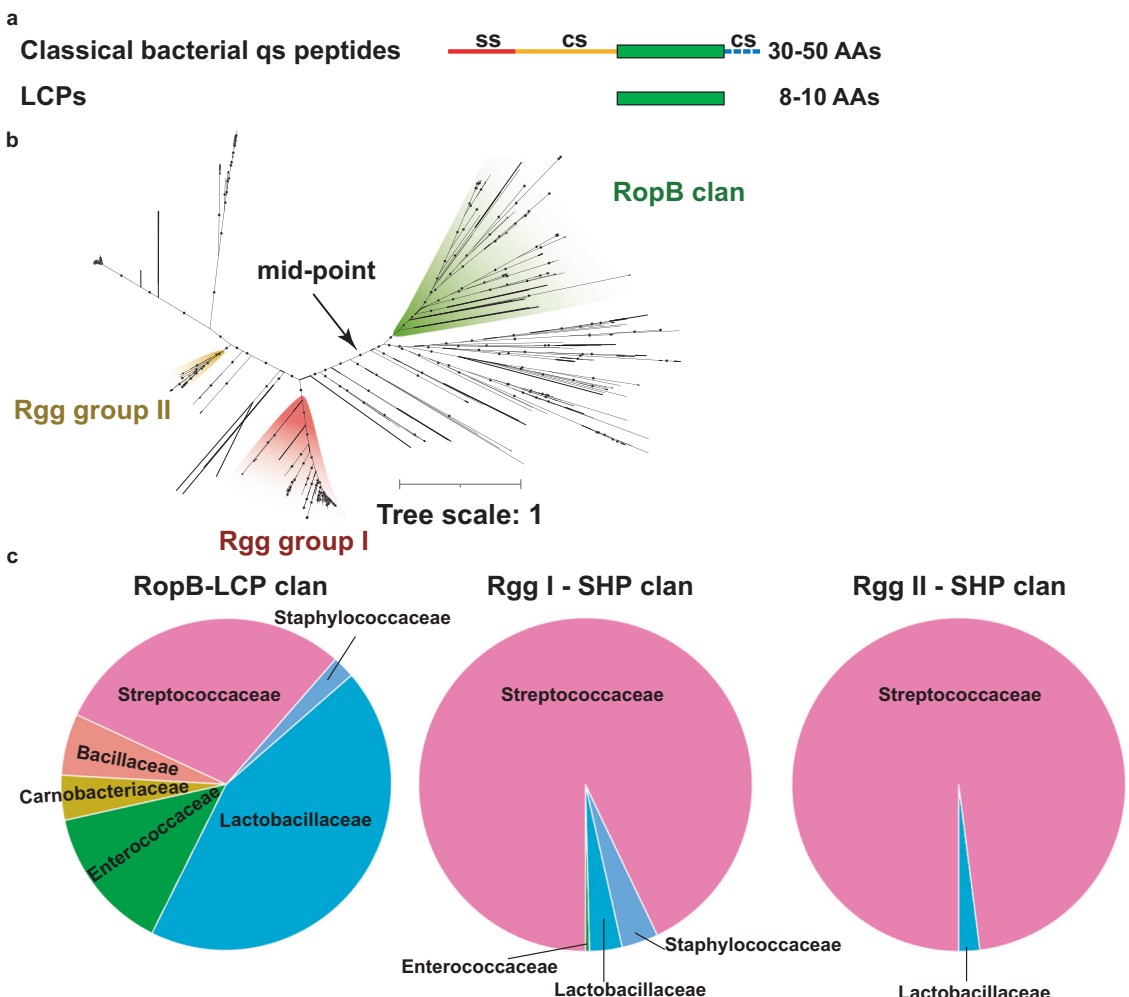

**Fig. 1 | Leaderless communication peptide (LCP) system is broadly distributed.**
**a** Schematics showing the amino acid sequence characteristics of classical bacterial qs propeptides and LCPs. Ss–secretion signal sequence; cs–protease cleavage sites; cs with dashed lines–indicate the presence of additional cleavage sites in a subset of propeptides. **b** The unrooted phylogenetic tree comprises 953 RopB homologs and was inferred from a trimmed alignment of 204 sites. Blues dots indicate branches supported by >90% bootstraps. Branch lengths are proportional to the expected number of substitutions per site, as indicated by the scale bar at the top left. Each highlighted clan corresponds to receptors for which the most likely translated adjacent usORF encodes a remarkable class of qs peptides. The two Rgg clans (red and yellow) are enriched in receptor-SHP propeptide pairs, while the RopB clan (green) is enriched in receptor-LCP pairs. **c** The distribution of RopB-LCP, Rgg I-SHPs, and Rgg II - SHPs systems in different bacterial taxa.

precluded their detection by gene annotation pipelines. Consequently, SIP remains as the only characterized LCP-based qs system, and the prevalence and distribution of LCPs in bacteria as well as their roles in bacterial pathophysiology and ecology remain unknown. To elucidate the full breadth of LCPs in bacteria, we designed a custom approach and scanned bacterial genomes for the presence of SIP-like LCPs. Here we report that LCPs are widespread among firmicutes encompassing a larger taxonomic diversity compared to other members of RRNPPA family regulators[12] and display a diverse set of communication codes with varying amino acid composition and lengths. We further show that a subset of the newly identified LCPs from distant bacterial species function as effective intercellular signals and control the expression of genes involved in diverse functions. Our findings point to the presence of a broadly distributed new class of qs signals and their potential roles in different aspects of bacterial pathophysiology.

## Results

### Leaderless communication peptide (LCP) system is broadly distributed

To assess the distribution of SIP-like LCP-based qs systems across bacteria, we employed a large-scale search strategy that includes: (i)

search for RopB homologs across a dataset of 129,001 bacterial genomes and 9,421 reference metagenomics-assembled genomes (MAGs), (ii) probing the genomic vicinity of RopB homologs for small ORFs with a preceding Shine-Dalgarno Ribosome-Binding-Site (RBS) motif that is indicative of their likely translation, (iii) identification of clans of RopB homologs for which the most likely translated adjacent sORF is ultrasmall and encodes a SIP-like LCP, and (iv) functional validation by assessing the regulatory activity of chosen subset of candidate LCPs.

A Blastp search of RopB against the target dataset resulted in 19,280 hits (sequence identity ≥25%, mutual length coverage ≥70%, Supplementary Data 1) encoded in 15,776 genomes and 39 MAGs distributed across 468 taxa. These 19280 hits correspond to 975 unique protein sequences that are predicted to harbor C-terminal tetra-tricopeptide repeat domain, a hallmark structural element responsible for peptide recognition by the RRNPPA family of receptors[7] (Supplementary Data 1). Importantly, 478 out of 975 RopB homologs (~49%) are flanked by at least one candidate sORF with a high confidence upstream RBS motif (Supplementary Data 1)[13,14].

To determine whether clans of RopB homologs are enriched for their association with LCPs, we inferred the phylogeny of RopB-like receptors and mapped the amino acid sequence of flanking sORF with

the strongest RBS motif on its corresponding receptor leaf of the tree (Fig. 1b, Supplementary Data 1). The two previously well-characterized clans of RopB homologs correspond to streptococcal Rgg regulators, which recognize adjacently encoded canonical propeptides that are post-translationally processed into mature short hydrophobic peptides (SHPs)[15,16]. Accordingly, the two Rgg clans appeared as two well-delineated clans in the RopB tree and were correctly mapped to SHP propeptides, which lends validation to our methodology to identify clans of qs systems (Fig. 1b and S1, Supplementary Data 1 and 2). Remarkably, RopB and its evolutionarily closest relatives represent a distinct clan of 183 receptors that were frequently associated with an ultrasmall peptide with amino acid composition similar to SIP (Fig. 1b, Supplementary Data 1 and 2). We term the new clan, RopB clan, and the receptors in RopB clan represent a new class of qs system that uses LCPs as signaling molecules.

The refined manual assessment of the candidate cognate LCP of each receptor within the RopB clan (Fig. 2 and Supplementary Data 2) revealed that the LCP system is broadly distributed in bacterial genomes of a large taxonomic diversity, spanning over *Streptococcaceae*, *Lactobacillaceae*, *Enterococcaceae*, *Carnobacteriaceae*, *Bacillaceae* and *Staphylococcaceae* families (Fig. 1c). Most of the candidate receptor-LCP pairs are encoded in the bacterial genomes, and a non-negligible number of LCP systems are present in either plasmids or integrative and conjugative elements (ICEs) (Fig. 3). However, receptor-LCP pairs were not detected in prophages or phages. The genetic elements encoding LCP-receptor pairs are found in various host-associated microbiomes, waste waters, and fermented food products. The LCP systems are prevalent in the genomes of several clinically relevant human pathogens such as *S. pyogenes* or *Enterococcus casseliflavus*, and in animal pathogens such as *S. porcinus* and *S. pseudoporcinus* (Fig. 3).

Most putative LCPs are 8 to 10 amino acids long with few exceptions such as LCPs from *Lysinibacillus sphaericus*, *Staphylococcus delphini*, *Enterococcus durans*, and *Granulicatella balaenopterae*. The predicted LCPs are highly hydrophobic and are mostly comprised of aliphatic and aromatic amino acids (Fig. 2 and S1). The peptides encode distinct communication codes with unique LCP amino acid sequences, suggesting that LCP-mediated communication is specific among closely related bacterial species or strains. Importantly, the characterized RopB-SIP system of *S. pyogenes* represents only a small subset of the identified LCP systems (Fig. 2), indicating the broader distribution of LCP systems and their potential unappreciated roles in bacterial pathophysiology.

The structure-guided multiple amino acid sequence alignment (MSA) analyses of RopB clan receptors indicate that the LCP-binding C-terminal domain (CTD) diversified faster than the N-terminal DNA-binding domain (DBD) (Figs. S2 and S3a)[7,11,17]. Among the structural elements of CTD, the α6 helix is highly conserved relative to the rest of CTD (Figs. S2 and S3a). The α6 helix constitutes a critical structural element for LCP binding as it forms the floor of the LCP binding pocket as well as engages in intramolecular interactions that provide the scaffold for the LCP-binding pocket of RopB (Supplementary Fig. 3b)[7,11,17]. In accordance with the suggested functional constraint on α6 helix to bind LCPs, the site-wise dN/dS ratio analyses indicate that α6 helix of candidate LCP receptors are under strong purifying selection against amino acid substitutions (Supplementary Fig. 3a). Similarly, the MSA analyses of 12 LCP-contacting amino acids of RopB with corresponding amino acids from candidate RopB-clan receptors suggest that these amino acids evolve slower and face stronger purifying selection compared to the reminder of CTD (Supplementary Fig. 4a, b). These observations are suggestive of faster diversification of TPR motifs likely due to their innate degeneracy[18,19] relative to LCP-receptor specificity diversification. In accordance with this, pairwise comparisons of evolutionary distances between LCP receptors, LCP-contacting residues in receptors, and candidate LCPs suggest that the amino acid sequences of LCP receptors diverged faster than the LCP-contacting residues and amino acid sequences of putative LCPs, and both LCPs and LCP-contacting amino acids in RopB clan receptors diverge at similar evolutionary rate (Supplementary Fig. 5). Collectively, these observations are suggestive of the function of RopB clan receptors and candidate LCPs as qs receptor-signal pairs.

To understand the co-evolution of receptors and candidate LCPs, we aligned the 12 LCP-contacting amino acids of RopB with similarly located amino acids from RopB-clan receptors and compared them with the physicochemical characteristics of the corresponding LCPs (Supplementary Fig. 4a, b). Consistent with the preponderance of aliphatic and aromatic amino acids in majority of candidate LCPs (Supplementary Fig. 1d), the peptide-contacting amino acids are relatively well conserved among RopB clan receptors (Supplementary Fig. 4). However, compared to RopB, the peptide-contacting residues of RopB clan receptors from *Ligilactobacillus muralis*, *Pediococcus acidilactici*, *Ligilactobacillus animalis*, and *Enterococcus casseliflavus* are distinct (Supplementary Fig. 4). Accordingly, the physicochemical properties of corresponding LCPs also deviate from the typical signature of LCPs as they contain charged, polar, and proline residues (Supplementary Fig. 4). These observations suggest a tropism of RopB clan receptors for SIP-like LCPs, however, divergence exists to achieve alternative LCP specificities through receptor-LCP co-evolution. Finally, analyses of the RopB clan receptor-LCP pairs from *Bacillus cereus* and *Lysinibacillus sphaericus* revealed an evolutionary feature that is suggestive of peptidase-mediated processing of some LCPs similar to canonical RRNPP propeptides (Supplementary Fig. 4). The predicted LCP binding sites and the corresponding candidate LCPs of *B. cereus* and *L. sphaericus* have identical amino acid composition. However, the LCP from *L. sphaericus* have an additional eight amino acids in their C-terminus compared to *B. cereus* LCP (Supplementary Fig. 4). This observation suggests that LCPs may exist in longer precursor forms and the C-terminal appendages may be involved in peptidase-mediated cleavage of precursor form to release mature LCPs.

Analyses of the genomic context of representative putative LCP systems showed that the candidate genes regulated by LCPs are predominantly in a divergent context relative to the receptor (Fig. 2) and belong to 3 major categories: biosynthetic gene clusters (BGCs) predicted to produce ribosomally synthesized and post-translationally modified peptide (RiPPs) as well as non-ribosomally synthesized antimicrobials, ABC-type transporters, and type VII secretion systems that are typically involved in the translocation of virulence factors (Fig. 3). These observations suggest a broader and more diverse role for LCP systems in bacterial pathogenesis, physiology, and microbial ecology.

## LCP in *S. salivarius* mediates gene regulation

To investigate whether the LCPs other than SIP also act as intercellular signals, we characterized the putative cytosolic receptor-LCP pair from *S. salivarius* (RopB$_{ss}$-LCP$_{ss}$). The *LCP$_{ss}$* is encoded in a megaplasmid and located downstream of *ropB$_{ss}$* and transcribed divergently (Fig. 4a). The *LCP$_{ss}$* encodes an eight amino acid hydrophobic peptide with a predicted amino acid sequence of MWLILLFL with no additional amino acids at either end (Fig. 4b). The genetic proximity of a 14-gene operon encoding a putative non-ribosomal peptide synthase biosynthesis gene cluster (*NRPS-BGC*) located immediately downstream of *LCP$_{ss}$* and transcribed convergently (Fig. 4a) suggest that *NRPS-BGC* is the regulatory target of the RopB$_{ss}$-LCP$_{ss}$ pathway.

In accordance with this, inactivation of *ropB$_{ss}$* or *LCP$_{ss}$* abrogated *NRPS-BGC* expression and *cis*-complementation of Δ*ropB$_{ss}$* and *LCP$_{ss}$** mutants with *ropB$_{ss}$* and *LCP$_{ss}$*, respectively, restored *NRPS-BGC* expression (Fig. 4c). Similarly, the addition of synthetic LCP$_{ss}$ containing the predicted amino acid sequence in native order (LCP$_{ss}$), not in scrambled order (SCRA), restored WT-like *NRPS-BGC* expression in the *LCP$_{ss}$** mutant (Fig. 4d). However, supplementation with synthetic

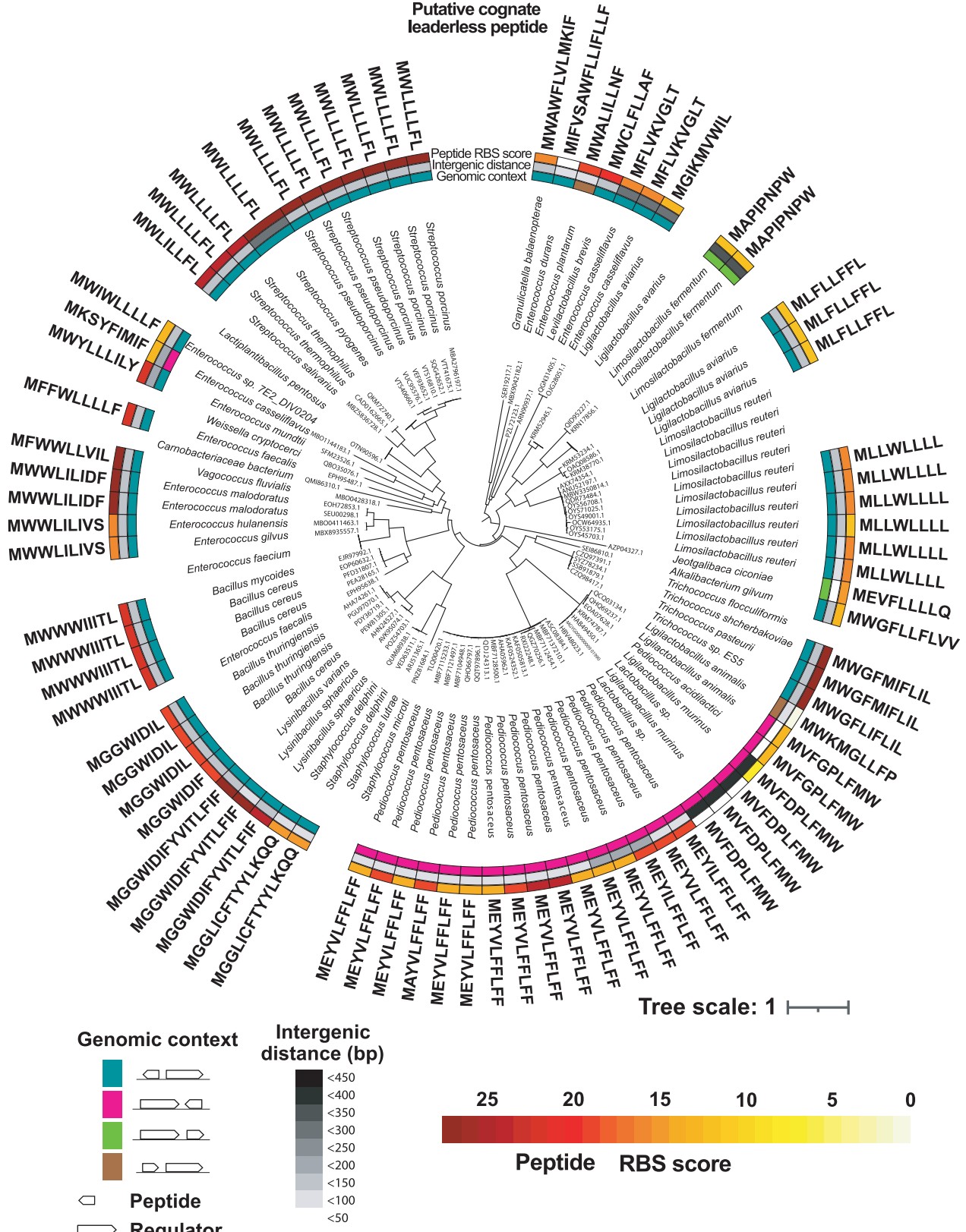

**Fig. 2 | Predicted LCPs encode diverse peptide communication codes.** The phylogenetic tree of the RopB clan, rooted at mid-point for visualization purpose, comprises 182 leaves and was inferred from a trimmed alignment of 229 sites. The label of a leaf corresponds to the NCBI protein id of the putative receptor, followed by the name of the encoding species. The most outer label in bold indicates the sequence of a putative LCP encoded in the genomic vicinity of a *ropB* homolog

(leaf) whenever detected. The three-color strips correspond to (i) the genomic orientation of the candidate receptor-LCP pair, (ii) the intergenic distance between the receptor and LCP ORFs and+ (iii) the confidence score, ranging from 1 to the 27 of the Shine-Dalgarno RBS motif identified upstream from the LCP's ORF (white meaning no RBS detected). Tips without labels correspond to collapsed receptors encoded by the same species and without LCP detected in the vicinity.

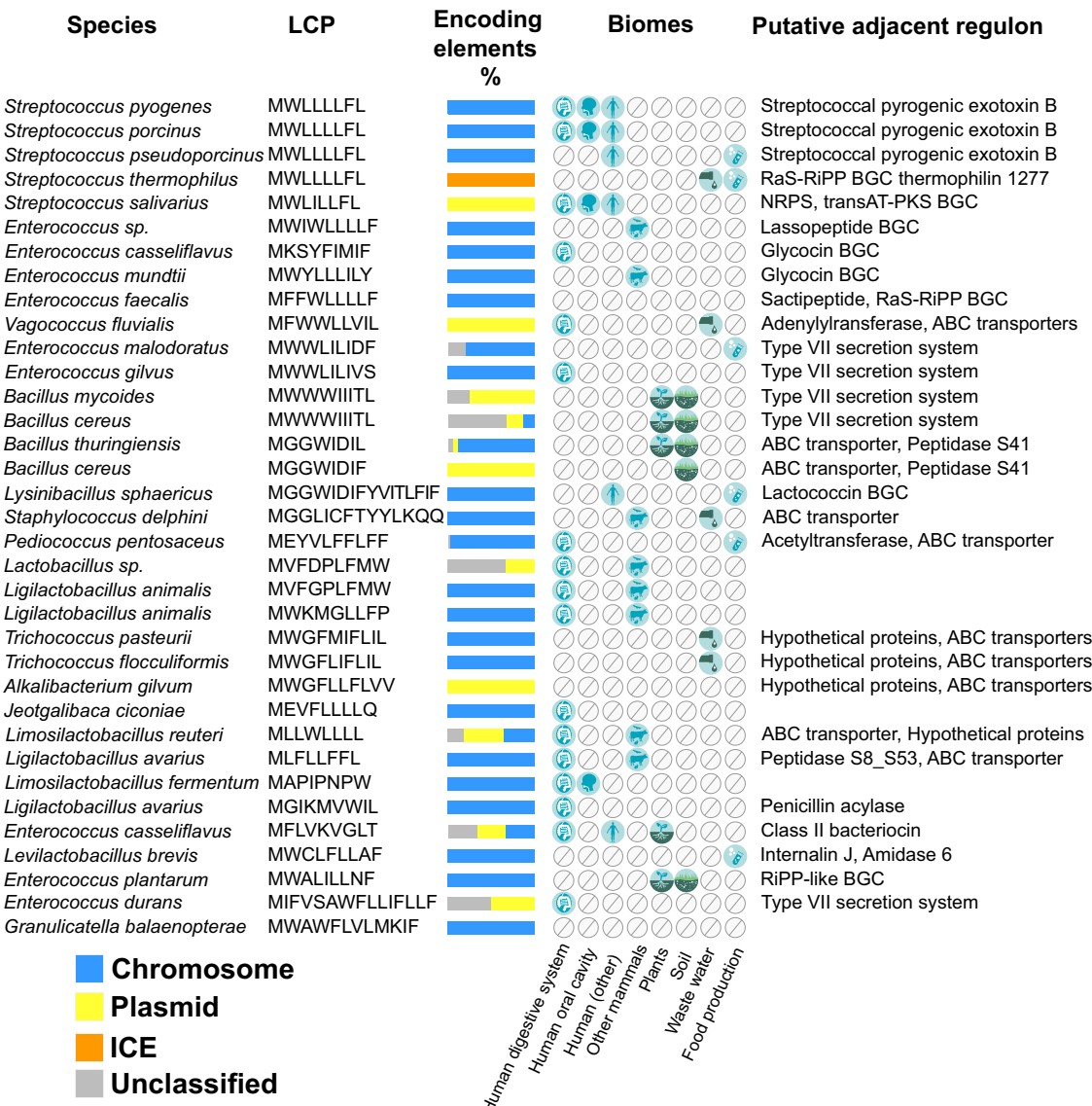

**Fig. 3 | Taxonomic, sequence, ecological and functional diversity of candidate LCP systems.** Each row depicts a representative of a group of receptor-LCP pairs with the same LCP sequence and the same species within the RopB clan. The stacked histogram highlights the proportion of receptor-LCP encoding elements in chromosomes, plasmids, ICEs or unclassified encoding contigs within a group. The matrix shows the biomes in which the LCP systems of a group were detected. The last column indicates the adjacently located putative target regulon controlled by each representative LCP system.

LCP$_{ss}$ containing staggering truncations at either N-terminal or C-terminal ends (Fig. 4d) failed to activate *NRPS-BGC* expression in the *LCP$_{ss}$** mutant, demonstrating that LCP$_{ss}$ encodes a mature LCP and lacks the hallmarks of canonical bacterial peptide signals (Fig. 4d). Furthermore, supplementation with even 20X molar excess of synthetic LCP$_{ss}$ failed to activate *NRPS-BGC* expression in Δ*ropB$_{ss}$* (Fig. 4d), indicating that LCP$_{ss}$ activity requires its cognate receptor RopB$_{ss}$. However, despite the absence of the secretion signal sequence, the LCP$_{ss}$ is secreted and reinternalized into the cytosol and acts as an intercellular signal. This was demonstrated by the presence of LCP$_{ss}$ associated regulatory activity only in the secreted component of peptide-producing strains (WT, *LCP$_{ss}$**::*LCP$_{ss}$*, and Δ*ropB$_{ss}$*::*ropB$_{ss}$*) and internalization of the exogenously added FITC-labeled synthetic LCP$_{ss}$ (Fig. 4e, f and Supplementary Fig. 6). Furthermore, inactivation of the canonical bacterial peptide import machinery oligopeptide permease (Δ*opp*) did not affect LCP$_{ss}$ import (Fig. 4f)[20], suggesting that unknown reimport mechanisms are involved in LCP$_{ss}$ import.

To elucidate the molecular mechanism of LCP$_{ss}$-mediated signaling, we investigated the sequence-specific recognition of LCP$_{ss}$ by cytosolic RopB$_{ss}$ by fluorescence polarization (FP) assay using FITC-labeled LCP$_{ss}$. RopB$_{ss}$ binds LCP$_{ss}$ with high affinity (K$_d$ ~8 nM) (Fig. 4g) and the pre-formed RopB$_{ss}$-FITC-LCP$_{ss}$ complex was disrupted only by unlabeled LCP$_{ss}$, not by non-specific SCRA (Fig. 4h), indicating that RopB$_{ss}$ recognition of LCP$_{ss}$ is sequence-specific. To explain the downstream consequences of RopB$_{ss}$-LCP$_{ss}$ interactions, we hypothesized that LCP$_{ss}$ facilitates RopB$_{ss}$ interactions with target promoters and promotes RopB$_{ss}$-dependent activation of *NRPS-BGC* expression. To map the operator sequences for RopB$_{ss}$ in *LCP$_{ss}$* and *NRPS-BGC* promoters, we performed electrophoretic mobility shift assays (EMSA) using different DNA fragments that span *LCP$_{ss}$* and *LCP$_{ss}$-NRPS-BGC* intergenic region (Supplementary Fig. 7). RopB$_{ss}$ bound only to a 43-bp fragment located immediately upstream of the putative −35 hexamer of the *LCP$_{ss}$* promoter and did not interact with the *NRPS-BGC* promoter (Fig. 4i and Supplementary Fig. 7). These results indicate that *LCP$_{ss}$* and *NRPS-BGC* are likely expressed as a polycistronic transcript and RopB$_{ss}$ binding site is located in the *LCP$_{ss}$* promoter (Supplementary Fig. 7). We further probed the 43-bp fragment for the presence of putative palindromes

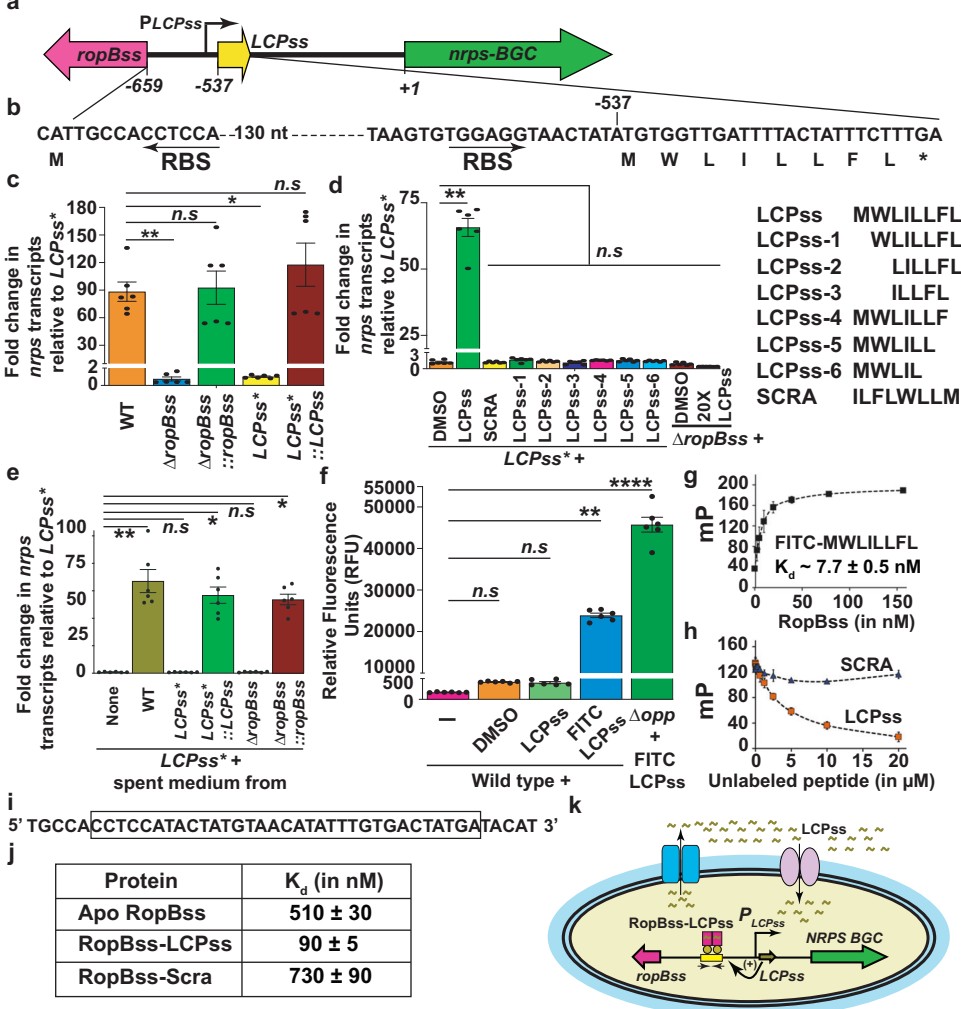

**Fig. 4 | Intercellular signaling and gene regulation by *S. salivarius* LCP.**
**a** Schematic representation of genetic elements in *S. salivarius* (ss) encoding *ropBss*, *LCPss*, and a biosynthetic gene cluster (BGC) encoding non-ribosomal peptide synthase (*nrps*). The divergently transcribed *ropBss* and *LCPss* along with predicted transcription start site (*PLCPss*, bent arrow) of *LCPss* are shown. The numbers below denote the nucleotide positions relative to the first nucleotide of the start codon of *nrps-BGC*. **b** The nucleotide sequence of the *ropBss-LCPss* intergenic region, coding sequence of *LCPss*, and corresponding predicted amino acid sequence of *LCPss* are shown. The ribosomal-binding sites (RBS) of LCPss and RopBss are denoted by arrows. **c** Analysis of *nrps* transcript levels in the indicated strains by qRT-PCR. **d** Addition of full-length synthetic LCPss activates *nrps* expression in LCPss inactivated mutant (*LCPss**). The amino acid sequences of the synthetic LCPss variants used are shown (right). **e** qRT-PCR based *nrps* transcript level analyses in *LCPss** mutant grown in spent medium from the indicated strains. **f** Cytosolic fluorescence corresponding to FITC-labeled LCPss indicative of the import of exogenous LCPss as assessed by fluorescence measurements. Unmodified or FITC-labeled LCPss was added at a final concentration of 1 μM to either WT or oligopeptide permease-inactivated mutant (Δ*opp*). After 30 min incubation, fluorescence in the clarified cell lysates was measured using excitation and emission wavelengths of 480 nm and 520 nm, respectively. **g** Analysis of the binding between purified RopBss and fluoresceinated LCPss by fluorescence polarization (FP) assay. **h** Ability of LCPss or SCRA peptide to compete with the FITC-labeled LCPss–RopBss complex for binding. A preformed RopB (250 nM)-labeled LCPss (10 nM) complex was challenged with the indicated unlabeled peptides. **i** Nucleotide sequence of the identified RopBss binding site in *LCPss* promoter used in the DNA-binding studies. **j** Summary of the affinity of different forms of RopBss to LCPss promoter as assessed by FP assays. **k** Proposed model for LCPss signaling. LCPss is produced, exported, and reinternalized into the cytosol. The recognition of LCPss by RopBss promotes high affinity interactions between RopBss and binding site in LCPss promoter, which leads to the upregulation of *LCPss* and *NRPS-BGC*. In (**c**, **d**, **e**, **f**), data are derived from three biological replicates and analyzed in duplicates. In (**g**, **h**), data are derived from three independent experiments. In (**c–h**), data graphed represent mean values ± s.e.m. *P* values in (**c**, **d**, **e**, **f**) were calculated by Kruskal-Wallis test. In (**c**), * - *P* = 0.0259, ** - *P* = 0.073. In (**d**), * - *P* = 0.014. In (**e**), * - *P* = 0.0114, ** - *P* = 0.0027. In (**f**), ** - *P* = 0.0016, **** - *P* < 0.0001. n.s not significant. Source data are provided as a Source Data file.

and found an inverted repeat with a 12 bp half site −4 bp spacer −12 bp half site motif that likely constitutes RopBss binding site (Supplementary Fig. 7e–f). However, the RopBss binding site differs from RopB-GAS binding site in several aspects including the motif arrangement, length, and nucleotide composition. The RopB-GAS binds to a palindrome with a 9 bp half site −7 bp spacer −9 bp half site motif (25 bp long)[10] compared to the 12 bp half site – 4 bp spacer – 12 bp half site motif (26 bp long) of RopBss. The half site of the palindrome in RopB-GAS binding site has a nucleotide composition of GTTACGTNT[10], which varies from RopBss binding site that has

nucleotide composition of ATGTAACATATT (Supplementary Fig. 7f). These findings indicate that the receptors recognize operator sequences of different length and nucleotide composition in the target promoters. However, consistent with the role of RopBss and RopB-GAS as transcription activators[10] and their likely role in the recruitment of RNA polymerase to defective promoters[21], the binding sites for both receptors are located upstream of and around the −35 region of LCP promoters.

To delineate the influence of LCPss on RopBss-promoter interactions, we assessed RopBss-DNA interactions in the presence and

absence of LCP$_{ss}$ by FP assay using FITC-labeled oligoduplex containing the identified RopB$_{ss}$ binding site (Fig. 4i). The addition of LCP$_{ss}$ resulted in high affinity interactions between RopB$_{ss}$ and the cognate DNA sequences (K$_d$ ~ 90 nM) compared to that of apo- or SCRA-bound RopB$_{ss}$ (K$_d$ ~ > 500 nM)(Fig. 4j and Supplementary Fig. 7g, h), indicating that LCP$_{ss}$ binding promotes high affinity interactions between RopB$_{ss}$ and LCP$_{ss}$ promoter. Based on these observations, we proposed a model for LCP$_{ss}$ signaling and LCP$_{ss}$-dependent transcription activation of *NRPS-BGC* by RopB$_{ss}$ (Fig. 4k).

**LCP system regulates streptococcal virulence factor production**
To assess the functionality of LCPs in other bacteria, we first assessed the regulatory activity of LCP from the swine pathogen *S. porcinus*. The amino acid sequence of *S. porcinus* LCP (LCP$_{sp}$) is identical to that of SIP (Fig. 5a, b). The coding region of *LCP$_{sp}$* is flanked upstream by *ropB* in the divergent direction and downstream by a gene encoding cysteine protease (*speB$_{sp}$*) that is transcribed convergently (Fig. 5a). Consistent with the role of LCP$_{sp}$ as an intercellular signal that controls *speB$_{sp}$* expression, supplementation of *S. porcinus* with synthetic LCP$_{sp}$ triggered early induction of *speB$_{sp}$* expression, while the non-cognate SCRA$_{sp}$ had no effect on gene regulation (Fig. 5c). Since the secreted cysteine protease SpeB is critical for the virulence of *S. pyogenes*[22,23], we reason that LCP$_{sp}$-mediated activation of *speB$_{sp}$* expression may impact the pathogenic traits of *S. porcinus*.

**LCPs from different clades of the RopB clan receptors phylogeny mediate intercellular communication and gene regulation**
To test whether LCP from non-streptococcal genus is functional, we characterized the regulatory activity of LCP from *Enterococcus malodoratus* (LCP$_{em}$) (Fig. 2). The LCP$_{em}$ is 9 amino acid long and its amino acid sequence is distinct from SIP (Fig. 5d, e). The *ropB$_{em}$* is divergently transcribed from *LCP$_{em}$*. However, unlike the other characterized LCPs (above), there are no convergently transcribed genes downstream of *LCP$_{em}$* (Fig. 5d). Instead, there are two genes encoding T7 secretion system-effector pair (T7SS) located downstream of *ropB$_{em}$* and transcribed convergently from *ropB$_{em}$*. Additionally, there are two hypothetical genes located downstream of *LCP$_{em}$* but transcribed divergently from *LCP$_{em}$* (Fig. 5d). Since the gene arrangement is distinct from other characterized LCP systems and regulatory influence of LCP$_{em}$ on these genes is unknown, we investigated the effect of synthetic LCP$_{em}$ on the expression of genes in both directions. Supplementation of *E. malodoratus* with LCP$_{em}$ induced only the expression of genes encoding *T7SS* and its effector and the induction was specific for LCP$_{em}$ (Fig. 5f). Contrarily, the LCP$_{em}$ had no influence on the expression profile of the two hypothetical genes located downstream of *LCP$_{em}$* (data not shown). These findings demonstrate that the LCP$_{em}$ that is dissimilar to SIP acts as an intercellular signal and controls the production of an *E. malodoratus* T7 secretion system/effector system.

To investigate the functionality of a LCP system from a more distant LCP system, we assessed the regulatory activity of LCP from *Limosilactobacillus reuteri* (LCP$_{lr}$) (Figs. 2, 6). Unlike other *L. reuteri* strains, the *L. reuteri* DSM32035 strain has a naturally occurring stop codon at amino acid position 4 of the putative LCP$_{lr}$ (Fig. 6b). The predicted untruncated full length LCP$_{lr}$ is 8 amino acid long (Fig. 6b, c) with the characteristic aliphatic and aromatic amino acid composition (Fig. 6a–c). The *ropB$_{lr}$* is divergently transcribed from *LCP$_{lr}$* (Fig. 6a). Two hypothetical genes encoding a putative ABC-type transporter are located downstream of *LCP$_{lr}$* and transcribed convergently from *LCP$_{lr}$* (Fig. 6a). Supplementation of synthetic LCP$_{lr}$ to the exponential growth of *L. reuteri* activated the naturally silent LCP pathway and induced the expression of genes encoding the ABC-type transporter. The induction was specific for LCP$_{lr}$ as the SCRA$_{lr}$ failed to activate the expression of ABC transporter (Fig. 6c). These results indicate that LCP from a distant species, LCP$_{lr}$, functions effectively as a qs signal and mediates gene regulation.

## Discussion
Here, we report the widespread prevalence of a new class of bacterial qs signals that remain largely uncharacterized for their roles in bacterial communication[10]. LCPs are unique bacterial qs peptides due to their synthesis from unannotated <30 base pairs usORFs and the absence of hallmarks of characterized bacterial peptide signals (Fig. 2). Notwithstanding the enormous advances in the understanding of bacterial quorum sensing over several decades[1,3,8], the unique features of LCPs prevented their identification and obscured the critical knowledge of the roles of an entire class of qs signals in bacterial group behaviors. To overcome this, we devised a tailor-made approach to detect LCPs and show that LCPs are prevalent among diverse bacterial species and encode an array of peptide communication codes (Figs. 1, 2, 3). Importantly, the discovery of LCPs indicates that usORFs as small as 24 base pairs can encode a functional bacterial qs signal thus rewriting the rules for the composition of bacterial qs peptides (Fig. 2)[10]. Many RRNPPA regulators are categorized as orphan receptors due to the absence of classical peptide ligands in their genetic vicinity[8,12,24,25]. In light of our findings, we reason that careful analyses of usORFs and their translated products in the genetic vicinity of orphan RRNPPA receptors may identify their cognate LCP ligands. In this regard, the methodology used here provides an effective tool to perform homology-based search to detect additional LCP systems. We anticipate that such analyses may uncover additional LCPs, expand LCP codes, elucidate new roles for LCPs in bacterial pathophysiology, and reveal an even broader distribution of LCPs than indicated by our study.

We demonstrate that, despite their presence as mature peptides without accessory sequences, several predicted LCPs act as intercellular signals and upregulate genes critical for various bacterial phenotypes (Figs. 4, 5, 6)[10]. Similarly, the genetic proximity of other uncharacterized LCPs to RopB homologs and the presence of a strong RBS upstream of their predicted start codons suggest that these usORFs also encode functional LCPs and participate in the coordination of bacterial group behaviors (Figs. 2, 3). Given the preponderance of LCPs in firmicutes, a prevalent phylum in human gut and oral microbiota[26,27], it will be interesting to explore the role of LCPs in communication among human-associated microbial communities, the functions they regulate, and their impact on human health.

On a broader note, the fundamental aspects of LCPs including their evolutionary origin and implications of usORFs encoding mature peptide signals, export and import mechanisms, regulatory mechanisms, genes and phenotypes regulated, and their contribution to microbial behavior remain unknown. Thus, our discovery of the broadly distributed LCPs may open new avenues of research and uncover exciting new knowledge regarding molecular mechanisms of qs signaling, gene regulation, and roles for LCPs in coordinating population-wide bacterial traits.

## Methods
### Distribution of putative signaling peptide-receptor pairs across the RopB-Rgg phylogeny
The complete genomes, chromosomes, and scaffolds of bacteria available on the NCBI Assembly database as of October 2021 were accessed using the following query string: ("Bacteria"[Organism] AND "latest genbank"[filter] AND ("complete genome"[filter] OR "scaffold level"[filter] OR "chromosome level"[filter])). The 448,667,348 protein sequences corresponding to these assemblies were downloaded from the NCBI using Genbank as a source database[28]. The metadata, protein sequences, genomes and annotations of the 9,246 high-quality species-representative MAGs of the four MAGs catalogs available on MGnify (cowrumen, human-gut, human-oral and marine) were downloaded from the following location on the Mgnify ftp server: http://ftp.ebi.ac.uk/pub/databases/metagenomics/mgnify_genomes[29]. The search for

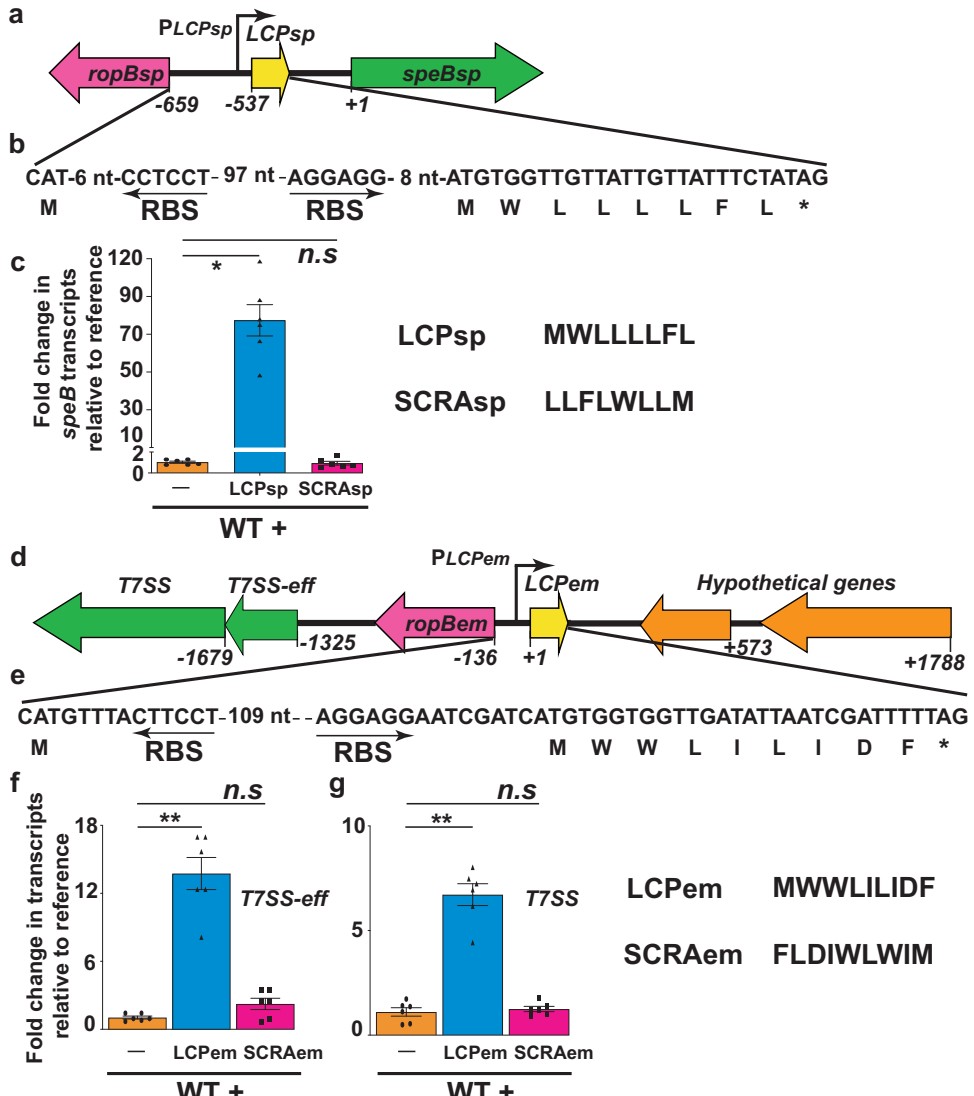

**Fig. 5 | Diverse LCPs mediate intercellular signaling in streptococcus and enterococcus. a** Schematic representation of genetic elements in *S. porcinus* (sp) encoding *ropB_sp*, *LCP_sp*, and secreted cysteine protease *speB_sp*. The *ropB_sp* and *LCP_sp* are divergently transcribed. The bent arrow above indicates the transcription start site of *LCP_sp*. The numbers below denote the nucleotide positions relative to the first nucleotide of the start codon of *speB_sp*. **b** The nucleotide sequence of the *ropB_sp*-*LCP_sp* intergenic region, coding sequence of LCP_sp, and corresponding predicted amino acid sequence of LCP_sp are shown. The ribosomal-binding sites (RBS) of LCP_sp and RopB_sp are marked by arrows. **c** Addition of synthetic LCP_sp causes early induction of *speB* expression in WT *S. porcinus*. *speB* transcript levels were assessed by qRT-PCR and the fold change in *speB* expression relative to unsupplemented growth (reference) is shown. **d** Schematics of genetic elements in *E. malodoratus* (em) encoding *ropB_em*, *LCP_em*, and genes encoding putative T7 secretion system (*T7SS*) and cognate effector (*T7SS-eff*). The bent arrow above

indicates the predicted transcription start site (*PLCP_em*) of *LCP_em*. The numbers below denote the nucleotide positions relative to the first nucleotide of the start codon of *LCP_em*. **e** The nucleotide sequence of the *ropB_em*-*LCP_em* intergenic region, coding sequence of LCP_em, and corresponding predicted amino acid sequence of LCP_em are shown. The ribosomal-binding sites of LCP_em and RopB_em are underlined. **f** Addition of synthetic LCP_em causes early induction of *T7SS* and *T7SS-eff* expression in WT *E. malodoratus*. Transcript levels were assessed by qRT-PCR and the fold change in *gene* expression relative to unsupplemented growth (reference) is shown. In **c**, **f**, data are derived from three biological replicates analyzed in duplicate and data graphed represent mean values ± s.e.m. *P* values in (**c**, **f**) were calculated by Kruskal-Wallis test. In (**c**), * - *P* = 0.0189, n.s not significant. In panel **f**, ** - *P* = 0.0015, n.s - not significant, whereas in (**g**), ** - *P* = 0.0049, n.s - not significant. Source data are provided as a Source Data file.

RopB homologs against these two target datasets was performed by BLASTP version 2.9.0+ with the option `–max_target_seqs` set to 1,000,000,000 to ensure that no RopB hits will be missing from the output file[30]. The output of BLASTP was filtered with the 25% identity and 70% mutual coverage cutoffs. These thresholds were chosen to capture a significant diversity of sequences while offering good guarantees on their expected function of receptor of small hydrophobic signaling peptides (irrespective of whether these peptides underwent post-translational cleavage (like SHPs) or not (like SIPs)), as these values correspond to the similarity observed between RopB and Rgg receptors in *S. pyogenes*. The presence of the

C-terminal peptide-interacting tetratricopeptide repeats of the RopB-Rgg superfamily in RopB homologs was determined by hmmsearch (HMMER version 3.3), using the TIGR01716 HMM profile from the TIGRFAM database as a query (E-value < 1E-5)[31]. The upstream region (−450 to +20 bp from start codon) and the downstream region (−20 to +450 bp from stop codon) of each of the *ropB* homologs were either extracted from the nucleotide fasta file of MAGs (MGnify) or downloaded from the NCBI (Genbank dataset) via https://eutils.ncbi.nlm.nih.gov/entrez/eutils/efetch.fcgi?db=nuccore&id =**<genomic_accession>**&seq_start =**<region_-start>**&seq_stop =**<region_end>**&rettype=fasta.

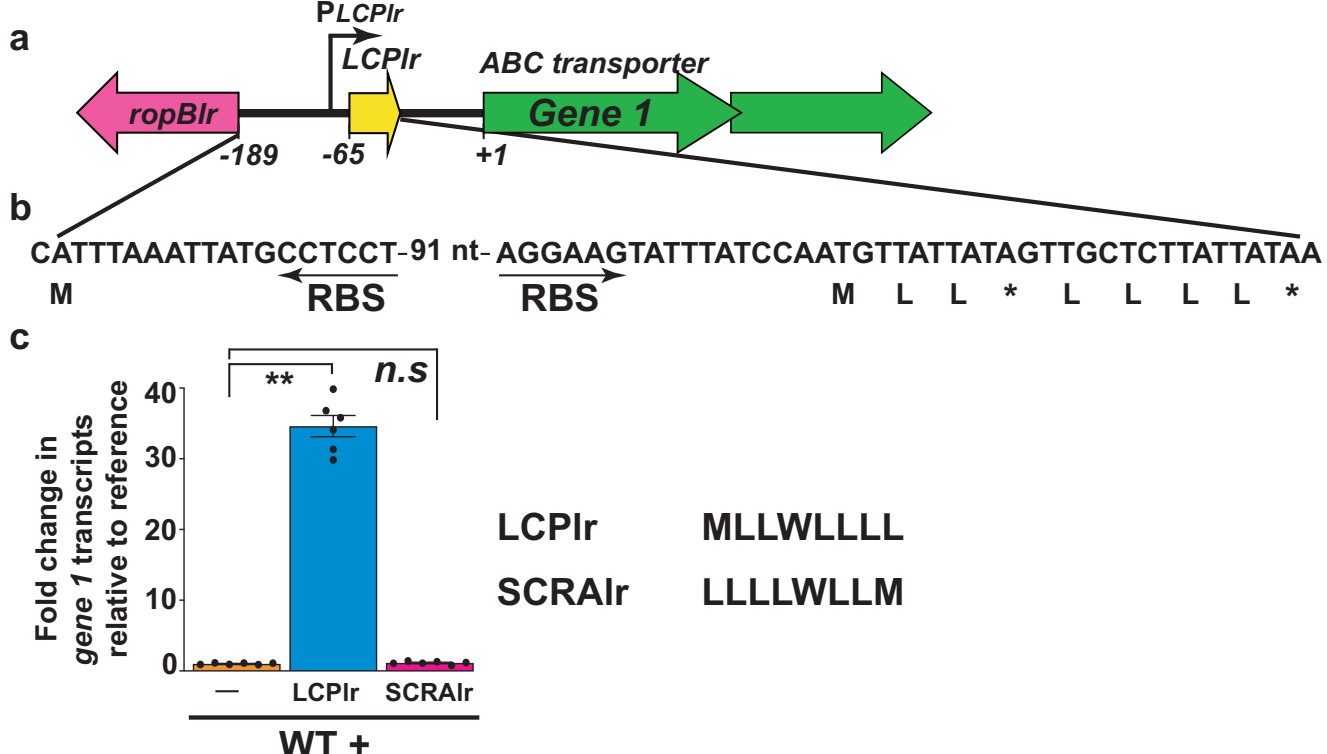

**Fig. 6 | LCP from a diverse species mediate intercellular signaling in *Limosi-lactobacillus reuteri*. a** Schematic representation of genetic elements in *L. reuteri* (*lr*) encoding *ropB_lr*, *LCP_lr*, ABC transporter (*Gene 1*). The *ropB_lr* and *LCP_lr* are divergently transcribed. The bent arrow above indicates the transcription start site of *LCP_lr*. The numbers below denote the nucleotide positions relative to the first nucleotide of the start codon of *gene 1*. **b** The nucleotide sequence of the *ropB_lr*-*LCP_lr* intergenic region, coding sequence of LCP_lr, and corresponding predicted amino acid sequence of LCP_lr are shown. The ribosomal-binding sites (RBS) of LCP_lr and RopB_lr are marked by arrows. **c** Addition of synthetic LCP_lr causes early induction of *gene 1* expression in WT *L. reuteri*. *Gene 1* transcript levels were assessed by qRT-PCR and the fold change in expression relative to unsupplemented growth (reference) is shown. In (**c**), data are derived from three biological replicates analyzed in duplicate and data graphed represent mean values ± s.e.m. *P* values in (**c**) were calculated by Kruskal-Wallis test. ** - *P* = 0.0024, n.s not significant. Source data are provided as a Source Data file.

The reverse complements of the two flanking regions of each *ropB* homolog were then generated for the minus strand to facilitate subsequent sORF calling and RBS motif detection. The Orfipy python software version 0.0.4 was launched against all these nucleotide flanking regions with the options `--strand f --min 18 --max 300 --start ATG` to select for any putative sORF encoding a micropeptide between 6 and 100 aa[32]. Nested micropeptides were then extracted based on the occurrence of an in frame ATG codon between the start and the stop codons of a longer ORF, if the resulting length was >= 6aa. The hierarchy of 27 regular expressions introduced by the Prodigal ORF-calling tool to detect Shine-Dalgarno RBS motifs was then applied against the upstream region of each detected ORF (−21 to −1bp from start codon)[13], consistent with the fact that >=90% of the protein coding genes encoded by Firmicutes are preceded by a Shine-Dalgarno RBS[14]. In a preliminary effort to assign a unique peptide to each RopB homolog in a stringent way, the neighboring micropeptide preceded by the RBS motif with the highest bin/score according to Prodigal was selected, providing that its score was above 13 and its usage across prokaryotes is greater than 3% according to Omotajo et al. (i.e bins 27, 24, 22, 19, 16 and 15)[14]. In case of an RBS score equality between peptides, a divergent context, and a small intergenic distance to ropB were prioritized. The phylogeny of RopB was then reconstructed as follows. First, RopB-like proteins were reduced to clusters of identical proteins using CD-HIT version 4.8.1 with the option `-c 1`[33]. Second, a multiple sequence alignment of the dereplicated RopB homologs was produced with mafft v7.453 with the options `-maxiterate 1000 -localpair` for high accuracy[34,35], and

trimmed with trimAl version 1.2.rev59 with the option `-automated 1` optimized for maximum likelihood phylogenetic reconstruction[36]. Third, the trimmed alignment of 174 sites was given as input to IQ-TREE version multicore version 1.6.12 to infer a maximum likelihood phylogenetic under the LG + G model with 1000 ultrafast bootstraps[37]. Finally, the topology of the tree and the peptide mapping onto the leaves of the tree were visualized using ITOL[38]. Remarkable clans of receptors were delimited based on the topology of the tree, and the amino-acid profile of mapped peptides. Receptor-peptide pairs within RopB and Rgg clans were further manually refined by carefully examining the list of all peptides encoded by a sORF in the vicinity of each representative *ropB* or *rgg* gene. SHPs were defined according to an amino-acid profile similar to that of the reference sequences listed in[15,16,39]. LCPs were identified based on a high RBS score, the consistency of mapped peptides across adjacent leaves of the tree, a genomic context preferably divergent relative to the receptor, a short length and an enrichment in aromatic and aliphatic residues (such as SIP).

**Computational characterization of candidate receptor-LCP pairs within the RopB clan**
The phylogeny of members of the RopB clan was inferred using mafft[34,35] and IQ-tree with the same parameters as cited above. The comparative analysis of amino-acid type enrichment and physicochemical properties between LCPs, SHPs and PrgX-like peptides were conducted as follows. First, mature SHP peptides and enterococcal PrgX-like peptides secreted via the same PptAB translocon as SHPs were retrieved from the Quorumpeps database[39]. The `aaComp` function

of the Peptides R package was used to create a matrix storing the abundance of each amino-acid type within each input quorum sensing peptide[40]. This matrix was used to create a heatmap with the ComplexHeatmap R package, with the quorum sensing peptides being clustered according their amino-acid type composition, using a Pearson correlation as the distance metric and the ward.D2 algorithm as the clustering method[41]. The 66 physico-chemical descriptors of each sequence were obtained by combining the output of the `blosumIndices`, `crucianiProperties`, `fasgaiVectors`, `kideraFactors`, `mswhimScores`, `ProtFP`, `stScales`, `tScales`, `vhseScales` and `zScales` functions of the Peptides R package. A PCA was then performed on this space of 66 dimensions with the `prcomp` function. The encoding element analysis was conducted as follows. First, the nucleotide fasta and the genbank files of each genomic accession / contig encoding a candidate receptor-LCP pair were fetched from the NCBI. Plasflow version 1.1 was run with default parameters against all these contigs in order to classify each contig according to one of these three categories: chromosome, plasmid or unclassified[42]. The genbank files were given as input to ICEscreen version 1.0.4 to identify regions corresponding to ICEs or IMEs within contigs[43]. Virsorter2 version 2.2.3 was launched with default parameters against the concatenated nucleotide fasta file and returned possible prophage regions within contigs[44]. The coordinates of ICEs, IMEs and prophages were then intersected with the coordinates of candidate LCP system(s) on corresponding contigs to define putative systems included within these mobile genetic elements. Because the provirus boundary detection algorithm in VirSorter2 purposely tends to overextend to host regions, the Phaster webtool was further used to confirm or infirm the prediction by Virsorter2 that a candidate LCP-based system is included within a prophage region (the few cases were all infirmed)[45]. The biome distribution analysis of LCP systems was performed as follows. First, the isolation_source tag was extracted from the genbank files of contigs encoding a candidate LCP system to obtain their environment of isolation. Next, a Blastp search of each receptor associated with a putative LCP was launched (sequence identity >= 90%, mutual coverage >= 90%) against the MGnify database of 623,796,864 clusters of >= 90% identical proteins: https://ftp.ebi.ac.uk/pub/databases/metagenomics/peptide_database/current_release/mgy_clusters.fa to obtain the biome of highly similar environmental sequences. The functional prediction of the target regulon of candidate LCP systems was conducted by running the Antismash webtool against the reference contig of each representative receptor-LCP pair[46]. Each BGC found right next to a receptor-LCP pair was assessed as the putative target regulon. When no adjacent BGC was identified, we carefully looked at the genomic orientation and the functional annotation of each neighboring genes. Of the two flanking regions, if one happened to be in a co-directional context relative to the LCP, this region was chosen as the putative target regulon.

## Co-evolution analysis of RopB-LCP pairs

The MSA of receptors from RopB clan paired with a candidate LCP was performed with mafft with the parameters –maxiterate 1000 –localpair for high accuracy. Coordinates of structural elements of RopB were extracted from AlphaFold prediction for the N-terminal DNA binding domain (AlphaFoldDB identifier O85731) and from the structurally resolved C-terminal LCP binding domain (PDB identifier: 6DQL). Structural elements and LCP-contacting residues were mapped onto the MSA, using RopB amino-acids as anchors. Sequences of LCP-contacting residues in RopB homologs were extracted from the previous MSA. Site-wise dN/dS ratios were derived as follows. First, coding sequences (CDS) of aligned receptors were fetched from the NCBI Genbank database. Second, the MSA was trimmed with trimAl to retain only sites with less than 20% gaps[36]. Third, a phylogenetic tree was derived from the trimmed alignment (with the same pipeline). Fourth, the protein MSA was converted into corresponding CDS MSA using

amino acid to codon mapping. Site-wise dN/dS were finally derived from the CDS MSA and the phylogenetic tree using HYPHY v2.5.51 with FEL-contrast method[47,48]. Comparisons of pairwise evolutionary distances between receptors, contacting residues and LCPs were derived as following. For each category (receptors, contact residues, LCPs), all vs all pairwise alignments were performed. Since sequences of LCPs and contacting residues are ultrasmall, exact Smith-Waterman alignment algorithm implemented in the fasta36 software v36.3.8i was used to ensure each alignment is the optimal one[49]. To derive an evolutionary distance metrics that is comparable between elements of different length (e.g., long receptors vs short LCPs / short contacting residues sequence), we used the following formula that normalize mutual bit scores between two sequences by the bit scores of their self-alignment: 1 – (bitscore A vs B + bitscore B vs A) / (bitscore A vs A + bitscore B vs B).

**Bacterial strains, plasmids, and growth conditions.** Bacterial strains and plasmids used in this study are listed in Supplementary Table 1. *S. salivarius* DS85_40B is a previously described strain isolated from the dietary supplements and food products whose genome has been fully sequenced[50]. *Escherichia coli* DH5α strain was used as the host for plasmid constructions and BL21(DE3) strain was used for recombinant protein overexpression. *S. salivarius, S. porcinus,* and *E. malodoratus* were grown routinely in Todd–Hewitt broth containing 0.2% (w/v) yeast extract (THY; DIFCO) at 37 °C. *Limosilactobacillus reuteri* was grown in DeMan, Rogosa and Sharpe (MRS) broth at 37 °C under microaerophilic growth conditions in a sealed chamber provided with anaerobe sachets (BD GasPak EZ Container Systems). When required, kanamycin was added to a final concentration of 100 μg/ml. Chloramphenicol was used at a final concentration of 15 μg/ml. All bacterial growth experiments were done in triplicate on three separate occasions for a total of nine replicates. Overnight cultures were inoculated into fresh media to achieve an initial absorption at 600 nm ($A_{600}$) of 0.03. Bacterial growth was monitored by measuring the absorption at 600 nm ($A_{600}$). The *E. coli* strain used for protein overexpression was grown in Luria Broth Miller (LB broth; Thermo Scientific).

**Construction of isogenic mutant and *cis*-complemented strains.** Isoallelic strains containing either single codon changes or inactivation of entire coding region were generated as previously described[51]. A DNA fragment with approximately 600 bp on either side of the coding region of interest was amplified using the primers listed in Supplementary Table 2 and cloned into the multi-cloning site of the temperature-sensitive plasmid pJL1055[52]. The resultant plasmids were introduced into *S. salivarius* by competence-based DNA uptake. Briefly, overnight *S. salivarius* growth was diluted in 0.3 ml of chemically defined medium[10] and incubated at 37 °C for 75 min. Subsequently, synthetic competence-stimulating (ComS) peptide with the amino acid sequence of LPYFAGCL and plasmid was added to the cells and incubated for 3 h at 37 °C. Cells were plated on agar plates containing appropriate antibiotics. The plasmid pJL1055 containing the intact genes were used to generate *S. salivarius* genetic revertant strains that has the gene of interest re-introduced into its original genetic locus. Colonies with plasmid incorporated into the *S. salivarius* megaplasmid were selected for subsequent plasmid curing. DNA sequencing was then performed to ensure that no spurious mutations were introduced.

**Construction of *opp$_{ss}$*-inactivated mutant strain.** Insertional inactivation of the *opp$_{ss}$* gene in WT *S. salivarius* was performed by methods described below. Briefly, a PCR fragment containing a spectinomycin resistance (spc) cassette with the fragment of gene to be deleted on either side was generated in a three-step PCR process. Subsequently, the plasmid with the spc gene disruption cassette was introduced into the parent strain by electroporation, and the gene was disrupted

through homologous recombination. The isogenic mutant strains were selected by growth on spectinomycin-containing medium. Inactivation of the gene was confirmed by DNA sequencing. Primers used for the construction of the isogenic $\Delta opp_{ss}$ mutant strain are listed in Supplementary Table 2.

**Transcript level analyses by qRT-PCR.** *S. salivarius*, *S. porcinus*, and *E. malodoratus* strains were grown in Todd-Hewitt Broth to late-exponential phase ($A_{600}$ ~2.0), early exponential phase ($A_{600}$ ~0.6), and mid-exponential phase ($A_{600}$ ~1.0) respectively. *L. reuteri* was grown in MRS broth to late-exponential phase (6 h post inoculation of the seed culture at 1:100 dilution). Cells were incubated with the corresponding LCP and cultures were immediately mixed with two volumes of ice-cold acetone and ethanol mixture and stored at −80 °C till use. Bacteria were harvested by centrifugation and RNA isolation and purification were performed using a RNeasy kit (Qiagen). Purified RNA was analyzed for quality and concentration with an Agilent 2100 Bioanalyzer. cDNA was synthesized from the purified RNA using High-Capacity cDNA Reverse Transcription Kit (Applied Biosystems) and qRT-PCR was performed with a Real-Time PCR Detection System (Bio-Rad). Comparison of transcript levels was done using $\Delta C_T$ method of analysis using the housekeeping genes mentioned in Supplementary Table 2.

**Culture supernatant swap assay.** To assess the presence or absence of $LCP_{ss}$ associated regulatory activity in the stationary growth phase culture supernatant of *S. salivarius* that induces *NRPS-BGC* expression, the indicated strains were grown to late-exponential phase ($A_{600}$ ~ 2.0). Cell-free culture supernatants were prepared by centrifugation and filtering through 0.22 μm membrane filter. The cell pellets of the wild type *S. salivarius* grown to mid-exponential growth phase ($A_{600}$ ~ 2.0) were resuspended in the secretome prepared from the mid-exponential growth phase of the indicated strains and incubated at 37 °C for 1 h. Transcript level analyses was performed by qRT-PCR as described above.

**Synthetic peptide addition assay.** Synthetic peptides of high purity (>90% purity) obtained from Peptide 2.0 (Chantilly, VA) were suspended in 100% DMSO to prepare a 10 mM stock solution. Stock solutions were aliquoted and stored at −20 °C until use. Working stocks were prepared by diluting in DMSO.

**Peptide reimport studies by microscopy.** To demonstrate the internalization of FITC-$LCP_{ss}$, *S. salivarius* cells were grown to late-exponential phase of growth ($A_{600}$ ~ 2.0), incubated with the indicated concentrations of unmodified or FITC-labeled $LCP_{ss}$ for 30 min at 37 °C. Cells were harvested by centrifugation, and washed three times with sterile PBS. Cells were fixed on the coverslip using 1% glutaraldehyde and 3% formaldehyde. The images were taken using a Nikon Eclipse TiN-STORM super resolution microscope equipped with iXon3 897 EM-CCD camera.

**Peptide reimport studies by fluorescence measurements.** To demonstrate the cytosolic internalization of exogenously added FITC-labeled $LCP_{ss}$, indicated *S. salivarius* strains were grown to late-exponential phase of growth ($A_{600}$ ~ 2.0) and incubated with either the indicated synthetic peptide or the carrier for the synthetic peptides (DMSO) for 30 min at 37 °C. Cells were harvested by centrifugation, washed three times with sterile PBS, and resuspended in equal volume of PBS. Cells were lysed by FastPrep-24 (MP Biomedicals) and lysates were clarified by centrifugation at 18,000 × *g* at 4 °C for 30 min. Samples were analyzed in 100 μl volume using an excitation and emission wavelengths of 490 and 520 nm, respectively. Readings were taken using a Biotek microplate reader (Biotek) and fluorescence measurements in relative fluorescence units were reported.

**Recombinant RopBss overexpression and purification.** The *ropBss* gene of *S. salivarius* was cloned into plasmid pET-28a and protein was overexpressed in *E. coli* strain BL21 (DE3) after induction with 1 mM IPTG at 25 °C for 16 h. Cell pellets were suspended in 50 ml of buffer A (20 mM Tris HCl pH 8.0, 100 mM NaCl and 1 mM Tris 2-carboxyethyl phosphine hydrochloride (TCEP)) supplemented with DNase I to a final concentration of 5 μg/ml. Cells were lysed by a Microfluidizer LM10 (Microfluidics) and cell debris was removed by centrifugation at 18,000 × *g* for 30 min. RopBss was purified by affinity chromatography using a Ni-NTA agarose column. The concentrated RopBss was further purified by size exclusion chromatography with a Superdex 200 G column and stored in buffer B (20 mM Tris HCl pH 8.0, 200 mM NaCl, 1 mM TCEP and 1 mM EDTA). The protein was purified to >95% homogeneity and concentrated to a final concentration of ~5 mg/ml.

**Electrophoretic mobility shift assays.** Probes containing different fragments of the $LCP_{ss}$ and $LCP_{ss}$-*NPRS-BGC* intergenic region were annealed by heating equimolar mixture of top and bottom strand oligonucleotides at 95 °C for 5 min followed by slow cooling to room temperature. Binding reactions were carried out in 20 μl volume of binding buffer (20 mM Tris pH 7.5, 0.8 M NaCl, 12% glycerol and 100 μM Zinc sulfate) containing 0.5 μM of oligoduplex and increasing concentrations of RopBss. After 15-min incubation at 37 °C, the reaction mixtures were resolved on a 10% native polyacrylamide gel supplemented with 5% glycerol (native polyacrylamide gel electrophoresis (PAGE)) for 90 min at 100 V at 4 °C in Tris Borate buffer with 5% glycerol. The gels were stained with ethidium bromide and analyzed on a BioRad Gel Electrophoresis Systems.

**Fluorescence polarization assay to assess protein-peptide and protein-DNA binding interactions.** Fluorescence polarization-based RopB$_{ss}$-ligand binding experiments were performed with a Biotek microplate reader (Biotek) using the intrinsic fluorescence of fluorescein labeled DNA or synthetic peptides. The polarization (P) of the labeled DNA or synthetic peptides increases as a function of protein binding, and equilibrium dissociation constants were determined from plots of millipolarization ($P \times 10^{-3}$) against protein concentration.

For RopB$_{ss}$–DNA-binding studies, 1 nM 5′-fluoresceinated oligoduplex in binding buffer (20 mM Tris–HCl pH 8.5, 200 mM NaCl, 1 mM TCEP and 25% DMSO) was titrated against increasing concentrations of purified RopB$_{ss}$ and the resulting change in polarization measured. Samples were excited at 490 nm and emission measured at 530 nm. The RopB$_{ss}$-peptide-binding studies were performed in a peptide-binding buffer composed of 20 mM potassium phosphate pH 6.0, 75 mM Nacl, 2% DMSO, 1 mM EDTA and 0.0005% Tween 20. All data were plotted using KaleidaGraph and the resulting plots were fitted with the equation $P = \{(P_{bound} - P_{free})[\text{protein}]/(K_D + [\text{protein}])\} + P_{free}$, where P is the polarization measured at a given protein concentration, $P_{free}$ is the initial polarization of the free ligand, $P_{bound}$ is the maximum polarization of specifically bound ligand and [protein] is the protein concentration. Nonlinear least squares analysis was used to determine $P_{bound}$, and $K_d$. The binding constant reported is the average value from at least three independent experimental measurements.

### Reporting summary
Further information on research design is available in the Nature Portfolio Reporting Summary linked to this article.

## Data availability
Data supporting the findings of this study are available in this article and its Supplementary Information files, or from the corresponding author upon request. Source data are provided with this paper.

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

## Acknowledgements

This work was supported in part by the National Institutes of Health grants R01 AI146048 and R01AI162748 to M.K. We thank Carmen Tartera for providing *S. salivarius* strain.

## Author contributions

S.A., E.H., H.D., N.M., Y.L., E.B., P.L., C.B., and M.K. designed and performed research; E.H., H.D., N.M., S.A., C.B., and M.K. analyzed data; S.A., E.H., H.D., C.B., and M.K. wrote the manuscript.

## Competing interests

The authors declare no competing interests.
