## [Peer Review File · Nature Communications]

The Leaderless Communication Peptide (LCP) class of quorum-sensing peptides is broadly distributed among FirmicutesReviewer #1 (Remarks to the Author):

The authors demonstrate that a class of QS peptides identified in 2017 could be widespread through the firmicutes. This data is interesting and relevant to the field. While novel, there are some issues with the data presented.

1. The supplemental tables are not well described so it is difficult to understand which peptides are shown in each table and each tab (ie. how the particular peptides were picked to show in which table). This made it difficult to go through the information.
2. I didn't see any statistical analysis or description of the stats used to evaluate any of the data in this paper. This is a huge issue that absolutely needs to be addressed before publication. Was there statistical difference in all of the qPCR shown, etc.?
3. The main novelty of the findings in this paper are that these peptides are much more widespread than previously thought. That is very interesting but then the authors chose to validate their findings using peptides from very closely related species *S. porcinius* and *S. salivarius*. While they did also choose one organism slightly further from the *Strep* (*Em*), this peptide was very similar in sequence and other characteristics (Figure 2) to the other peptides.

To increase the novelty of the findings, it would be important to show that a more distantly related organism with a different type of peptide/regulator pair predicted by their system also works. In addition, statistics would be necessary before any evaluation of the work could be done.

Reviewer #2 (Remarks to the Author):

In this bioinformatic, genetic, and biochemical study, Huang and collaborators characterize the distribution of a new communication system within the RRNPAA family of quorum sensing (QS). RRNPAA is the most abundant QS system family in Bacillota using as communication molecule short peptides (5-10 aa) that are generated from pro-peptides (40-100 aa) in an export-import process. In contrast, this new group analyzed in the manuscript is characterized by the signaling peptide already being produced in its mature form and therefore not undergoing the export-import process to be functional. As a result, the signaling peptides lack export and processing sequences and have been termed by the authors as leaderless communication peptides (LCP). Based on the data obtained from the characterization of RopB from *Streptococcus pyogenes*, the prototype receptor for this new family, previously carried out by the same group (Do et al., PNAS, 2017; PMID: 28923955), the authors analyze in silico the distribution of this new class of RRNPAA receptors and find that they are widely distributed in Bacillota, present not only in bacterial genomes but also in different mobile genetic elements (plasmids and ICEs), but not in phages. The authors show that this wide distribution leads to variability in signaling peptides (size and sequence) and in the genomic organization of the receptor-signaling peptide. The functional characterization of three members of this group confirms a similar mechanism of action. Undoubtedly, the results presented are of interest as they reveal the importance and distribution of this new group within the RRNPAA family, of which only RopB was previously known. However, I believe that the results in their current state, as important as they are, are not such a significant advancement compared to the previous results published by the group to justify their publication in a journal with the high standards of *Nature Communications*. To justify the publication in this journal, the authors must take a step further and analyze some of the points highlighted in the current manuscript in more depth. Otherwise, the advancement from their previous work is low, as evidenced in some of the figures, which are basically identical to those published in the 2017 paper (for example, Fig. 3k in the current paper and Fig. 6E in Do et al., 2017). I would be keen to recommend acceptance of this paper after comments below are addressed.

Comments:

- i) The in-silico analysis demonstrating the distribution and variability of this new group is a great contribution of this work, which not only highlights that these systems are widely distributed but also exhibit different genomic organizations. Undoubtedly, this must have required a significant effort, so I miss a more in-depth analysis, reducing it to a descriptive presentation of the results.

It is possible that the authors present these results in future publications, but I believe that to justify publication in a journal such as Nature Communications, a more thorough analysis should be provided. Specifically, I miss an analysis of the variability of the signaling peptides with respect to their receptors, and within these, the variability between the DNA-binding and peptide-binding domains, correlating these data with the receptor-peptide genomic organization and/or whether they belong to a mobile genetic element or bacteria genome.

ii) The authors do an excellent job delimiting the operator to which the receptor binds and bounding it to 43 base pairs. Knowing the quaternary organization of this receptor, as the authors have determined its three-dimensional structure in its apo form (PDB 5DL2; PMID: 26714274) and bound to the regulatory peptide (PDB 6DQL; PMID: 31197146), it is surprising that they do not locate and characterize putative binding palindromes. This data would be of great importance in locating putative binding sites for other members of this family, especially considering the variable genomic organization highlighted in the manuscript. This point should be addressed.

iii) To study the functionality of this new class of receptors, the authors analyzed three systems. Two of them belonging to different Streptococcus species are very similar, both in their organization and in the sequence of the signaling peptides, to the previously analyzed from *S. pyogenes*, so it could basically be a confirmation of the previous data. The third system belonging to *Enterococcus malodoratus* does show variability in the peptide, although the receptor-peptide genomic organization (divergent) is identical, only varying the orientation of downstream genes from the peptide. It is not clear to this reviewer why these systems were selected, and not others that are more distantly related (peptide-receptor in convergent or sequential organization) that, in correlation with the previous points, could have provided greater mechanistic information on the conservation or variability of this new group.

Reviewer #3 (Remarks to the Author):

In this work, the authors describe the identification of a new class of peptide-based quorum sensing signals, termed leaderless communication peptide (LCP), that were completely overlooked until very recently, and that are broadly distributed across Gram-positive genera. The authors first conducted a comprehensive genome mining analysis of 129,000 bacterial genomes and developed specific tools to analyze the data and identify potential novel LCPs, leading to the identification of 183 potential new LCP-based signaling systems. The authors then went on to validate the functionality of three representative systems from two different genera, Streptococcus and Enterococcus. The authors were able to validate the functionality of these signaling systems, characterize the activity of these LCPs as intercellular signals that are secreted and get reinternalized into the cytosol, and exhibit that these circuitries regulate diverse bacterial phenotypes and behaviors. Overall, this is an excellent contribution to the fields of quorum sensing and bacterial communication with far reaching implications to how signaling molecules are viewed and evaluated. I believe that this work is novel, very significant, and should be of interest to the broader scientific community. Furthermore, I found this work to be cutting-edge and meticulously executed, and therefore recommend it to be published in Nat. Commun. after the authors address a few minor issues:

- Generally, although the paper is well-written, it has some language issues throughout. I recommend that the authors carefully review the paper and correct these issues, or utilize a professional service to correct the text.
- Additionally, the text contains a few typos that need to be corrected. For example, on page 10, line 220, "...acts an intercellular..." should be "... acts as intercellular...".
- Also, in the Figure legend of Figure S3 in the SI, "...whereas varying concentrations LCPss were used." should be "...whereas varying concentrations of FITC-LCPss were used."

RESPONSE TO REVIEWER'S COMMENTS:

We thank the reviewers for their time and detailed comments that improved the clarity and depth of the revised manuscript. Below are our point-by-point responses to reviewers comments.

MAJOR REVISIONS:

1. Need to provide functional (experimental) validation of at least one more distantly related system (reviewers 1 and 2)

The strain availability from a public biological resource center (ATCC or DSMZ) and ability to grow them under aerobic conditions limited the functional characterization to streptococcal and enterococcal LCP systems in the original manuscript. The vast majority of the remaining bacterial species encoding candidate LCPs only grow anaerobically and require specific instruments and growth conditions to cultivate and perform peptide addition experiments.

However, to address reviewer's comments about characterizing a distantly related system (**Fig. 1 below; Figure 6 in the revised manuscript**), we tested the gene regulatory activity of candidate LCP system from *Limosilactobacillus reuteri*.

We show that the addition of synthetic peptide containing the amino acid sequence of LCP from *L. reuteri* induced the expression of gene located downstream of LCP in a sequence-specific fashion (**Fig. 6 and lines 296-308 in the revised manuscript**). These results show that LCP from a distantly related system is functional.

We have also obtained and attempted to characterize candidate LCP systems from strict anaerobes such as *Ligilactobacillus aviarius*, *Lysinibacillus sphaericus*, *Pediococcus pentosaceus* and *Trichococcus pasteurii* strains. However, unlike *Limosilactobacillus reuteri*, none of the above-mentioned strains grew in microaerophilic growth conditions (anaerobic jar

Figure 1. Illustration showing the candidate LCP systems characterized in this study. The LCP systems characterized in the original manuscript are highlighted in pink, whereas the LCP system validated in the revised draft is shaded in green.

with packs) and require strict anaerobic incubators. Thus, despite our significant efforts, we were unable to grow anaerobes and assess the functionality of the putative LCP systems in those strains.

Nevertheless, we reason that our findings provide a catalog of previously unknown class of bacterial qs signals, LCPs, to the community. Importantly, the validation of 4 different LCP systems from distinct bacteria that are distributed into 3 different clades of RopB phylogeny (**Figs. 4, 5, and 6 in the revised manuscript**) lend confidence to pursue the characterization of the newly identified LCP systems in anaerobes by laboratories equipped to cultivate anaerobic bacteria.

2. In response to reviewer 2 comments, we have performed in-depth analyses of the relationship between different LCPs and their receptors and its relevance to structural elements of the receptor and their genomic organization. We have included the details in the revised manuscript (**Lines 142 - 185 in the revised manuscript, Supplementary figs S2-S5**).

3. In accordance with reviewer 1 comments, we have performed statistical analyses and validated the statistical differences between LCP and SCRA-supplemented samples in the upregulation of gene expression (**Figures 4, 5, and new figure 6 in the revised manuscript**). We have also included the details of statistical analyses methods in the figure legends.

4. As noted by reviewer 3, we have carefully edited the manuscript to avoid typos.

5. We have provided our point-by-point response to reviewer's comments below.

REVIEWER COMMENTS:

Reviewer #1 (Remarks to the Author):

The authors demonstrate that a class of QS peptides identified in 2017 could be widespread through the firmicutes. This data is interesting and relevant to the field. While novel, there are some issues with the data presented.

1. The supplemental tables are not well described so it is difficult to understand which peptides are shown in each table and each tab (ie. how the particular peptides were picked to show in which table). This made it difficult to go through the information.

RESPONSE: We have revised the supplementary table S1 to provide more clarity and facilitate the reader's ability to go through the information. Specifically, we grouped the columns in three categories: MAIN INFO, GENOME DETAILS, AND RECEPTOR_DETAILS. Furthermore, we highlighted the peptide picked-up by the automated procedure in yellow in the first tab of Table S1 that lists all flanking putative sORFs associated with receptors.

2. I didn't see any statistical analysis or description of the stats used to evaluate any of the data in this paper. This is a huge issue that absolutely needs to be addressed before publication. *Was there statistical difference in all of the qPCR shown, etc.?*

RESPONSE: We have performed statistical analyses of the data to demonstrate significant differences between LCP supplemented and control samples in LCP-specific induction of target genes (**Figures 4, 5, and new figure 6 in the revised manuscript**). We have provided the details of statistical analyses in the figure legends.

3. The main novelty of the findings in this paper are that these peptides are much more widespread than previously thought. That is very interesting but then the authors chose to validate their findings using peptides from very closely related species *S. porcinus* and *S. salivarius*. While they did also choose one organism slightly further from the *Strep* (*Em*), this peptide was very similar in sequence and other characteristics (Figure 2) to the other peptides.

RESPONSE: We chose the streptococcal and enterococcal LCP systems for functional characterization due to the strain availability from a public biological resource (ATCC or DSMZ) and our ability to grow them under aerobic conditions. The vast majority of the remaining bacterial species only grow anaerobically and require specific instruments/growth conditions to cultivate and perform peptide addition experiments. However, in response to reviewer's comments, we have characterized the LCP system from anaerobic bacteria *L. reuteri* and provided experimental validation that the system is functional in a bacterial species distant from the group of bacterial species characterized in the original manuscript (**also see our # 1 comment on the top – Major revisions**).

To increase the novelty of the findings, **it would be important to show that a more distantly related organism with a different type of peptide/regulator pair predicted by their system also works.** In addition, **statistics would be necessary before any evaluation of the work could be done.**

RESPONSE: As noted above, we have performed both to address reviewer's comments.

Reviewer #2 (Remarks to the Author):

In this bioinformatic, genetic, and biochemical study, Huang and collaborators characterize the distribution of a new communication system within the RRNPAA family of quorum sensing (QS). RRNPAA is the most abundant QS system family in Bacillota using as communication molecule short peptides (5-10 aa) that are generated from pro-peptides (40-100 aa) in an export-import process. In contrast, this new group analyzed in the manuscript is characterized by the signaling peptide already being produced in its mature form and therefore not undergoing the export-import process to be functional. As a result, the signaling peptides lack export and processing sequences and have been termed by the authors as leaderless communication peptides (LCP). Based on the data

obtained from the characterization of RopB from *Streptococcus pyogenes*, the prototype receptor for this new family, previously carried out by the same group (Do et al., PNAS, 2017; PMID: 28923955), the authors analyze in silico the distribution of this new class of RRNPPA receptors and find that they are widely distributed in Bacillota, present not only in bacterial genomes but also in different mobile genetic elements (plasmids and ICEs), but not in phages. The authors show that this wide distribution leads to variability in signaling peptides (size and sequence) and in the genomic organization of the receptor-signaling peptide. The functional characterization of three members of this group confirms a similar mechanism of action. Undoubtedly, *the results presented are of interest as they reveal the importance and distribution of this new group within the RRRNPPA family, of which only RopB was previously known.* However, I believe that the results in their current state, as important as they are, are not such a significant advancement compared to the previous results published by the group to justify their publication in a journal with the high standards of *Nature Communications*. To justify the publication in this journal, the authors must take a step further and analyze some of the points highlighted in the current manuscript in more depth. Otherwise, the advancement from their previous work is low, as evidenced in some of the figures, which are basically identical to those published in the 2017 paper (for example, Fig. 3k in the current paper and Fig. 6E in Do et al., 2017). I would be keen to recommend acceptance of this paper after comments below are addressed.

Comments:

i) The in-silico analysis demonstrating the distribution and variability of this new group is a great contribution of this work, which not only highlights that these systems are widely distributed but also exhibit different genomic organizations. Undoubtedly, this must have required a significant effort, so I miss a more in-depth analysis, reducing it to a descriptive presentation of the results. It is possible that the authors present these results in future publications, but I believe that to justify publication in a journal such as *Nature Communications*, a more thorough analysis should be provided. Specifically, I miss an analysis of the variability of the signaling peptides with respect to their receptors, and within these, the variability between the DNA-binding and peptide-binding domains, correlating these data with the receptor-peptide genomic organization and/or whether they belong to a mobile genetic element or bacteria genome.

RESPONSE: To address the reviewer's comment, we have performed in-depth analyses of RopB clan receptors and their predicted cognate peptides. Our analyses revealed that i) among the structural elements of the receptors, the C-terminal peptide-binding domain (CTD) diversified faster than the N-terminal DNA-binding domain (NTD).

ii) the following observations support the co-evolution of receptors and LCPs:

- a. The α helix 6 of CTD forms the floor of the LCP-binding pocket and engages in intramolecular interactions critical for the LCP-binding pocket scaffold in RopB. Consistent with this critical role, the amino acid sequence of α helix 6 of CTD is highly conserved and under strong purifying selection against amino acid

substitutions (**Figs. S2 and S3** in the revised manuscript). These observations suggest that α helix 6 of is likely functionally constrained to recognize a LCP.

- b. Similarly, the LCP-contacting amino acids of RopB clan receptors are under strong purifying selection relative to majority of CTD residues (**Fig. S4** in the revised manuscript).
- c. Comparisons of pairwise evolutionary distances between receptors, LCP-contacting residues, and amino acid sequence of candidate LCPs indicate that LCP non-contacting residues diverged faster than LCP-contacting residues and LCP sequences. These analyses also suggest that LCP contacting residues and LCPs diverge at the same evolutionary rate (**Fig. S5** in the revised manuscript).

Collectively, these findings suggest the co-evolution of RopB clan receptors and LCPs, and their likely function in quorum sensing pathways.

iii) to understand the co-evolution of receptors and peptides, we aligned the peptide-contacting residues of RopB with other receptors and compared them with the physicochemical characteristics of the corresponding LCPs. We found that peptide-contacting residues are relatively well conserved among receptors and consequently, the corresponding LCPs are characterized by aliphatic and aromatic residues. However, we also noted exceptions to this observation as receptors from *Ligilactobacillus muralis*, *Pediococcus acidilactici*, *Ligilactobacillus animalis*, and *Enterococcus casseliflavus* have unique composition of peptide-contacting residues compared to RopB. Consistent with this, the putative LCPs have charged and polar amino acids, suggesting that LCPs with physicochemical signatures different than SIP-like LCPs may exist and these receptor-LCP pairs may have coevolved to achieve specificity (**Fig. S4** in the revised manuscript).

iv) Finally, we noted that the receptors from *Bacillus cereus* and *Lysinibacillus sphaericus* have identical peptide-contacting residues. However, the corresponding LCP of *L. sphaericus* is predicted to be longer than that of *B. cereus*. These findings suggest that despite the lack of secretion signal sequences, LCPs may also exist in longer form and require processing by proteases to release the mature LCP (**Fig. S4** in the revised manuscript).

We discussed these observations and their potential implications in the revised draft (**Lines 142 - 185 in the revised manuscript, Supplementary figs S2-S5**).

ii) The authors do an excellent job delimiting the operator to which the receptor binds and bounding it to 43 base pairs. Knowing the quaternary organization of this receptor, as the authors have determined its three-dimensional structure in its apo form (PDB 5DL2; PMID: 26714274) and bound to the regulatory peptide (PDB 6DQL; PMID: 31197146), it is surprising that they do not locate and characterize putative binding palindromes. This data would be of great importance in locating putative binding sites for other members of this family, especially considering the variable genomic organization highlighted in the manuscript. This point should be addressed.

RESPONSE: In accordance with the reviewer's comment, we probed the identified 43-bp binding site for the presence of putative palindromes. We found an inverted repeat with two mismatches and a 12 bp half site – 4 bp spacer – 12 bp half site motif that likely

constitutes RopB_{ss} binding site. However, the RopB_{ss} binding site differs from RopB-GAS binding site in several aspects including the motif arrangement, length, and nucleotide composition. The RopB-GAS has a 9 bp half site – 7 bp spacer – 9 bp half site motif (25 bp long), whereas RopB_{ss} binding site has a 12 bp half site – 4 bp spacer – 12 bp half site motif (26 bp long). The half site of the repeat of RopB-GAS has a composition of GTTACGTNT (N – A or G), which varies from RopB_{ss} binding site that has nucleotide composition of ATGTAACATATT. We marked the identified putative RopB_{ss} binding palindrome in **Fig. S6e-f**. These findings indicate that the receptors recognize different motifs in target promoters but the motifs are located upstream of LCP_{ss} and around -35 region of LCP_{ss} promoter. We anticipate that these findings may aid the identification of DNA binding sites for other LCP receptors.

We have incorporated these findings in the revised draft (**Lines 240 - 253 in the revised manuscript**).

iii) To study the functionality of this new class of receptors, the authors analyzed three systems. Two of them belonging to different *Streptococcus* species are very similar, both in their organization and in the sequence of the signaling peptides, to the previously analyzed from *S. pyogenes*, so it could basically be a confirmation of the previous data. The third system belonging to *Enterococcus malodoratus* does show variability in the peptide, although the receptor-peptide genomic organization (divergent) is identical, only varying the orientation of downstream genes from the peptide. It is not clear to this reviewer why these systems were selected, and not others that are more distantly related (peptide-receptor in convergent or sequential organization) that, in correlation with the previous points, could have provided greater mechanistic information on the conservation or variability of this new group.

RESPONSE: Our choice of candidate LCP systems for functional characterization was dictated by the strain availability from public biological resource centers (ATCC or DSMZ) and our ability to grow them under aerobic conditions. Except for the streptococcal and enterococcal strains, the vast majority of the remaining bacterial species only grow anaerobically and require specific instruments/growth conditions to cultivate and perform peptide addition experiments. However, in response to reviewer's comments, we have characterized the LCP system from anaerobic bacteria *L. reuteri* that we were able to grow under microaerophilic conditions and provided experimental validation that the system is functional in a bacterial species relatively distant from the group of bacterial species characterized in the original manuscript (**also see our # 1 comment on the top – Major revisions**).

Reviewer #3 (Remarks to the Author):

In this work, the authors describe the identification of a new class of peptide-based quorum sensing signals, termed leaderless communication peptide (LCP), that were completely overlooked until very recently, and that are broadly distributed across Gram-positive genera. The authors first conducted a comprehensive genome mining analysis of 129,000 bacterial genomes and developed specific tools to analyze the data and identify potential novel LCPs, leading to the identification of 183 potential new LCP-based

signaling systems. The authors then went on to validate the functionality of three representative systems from two different genera, Streptococcus and Enterococcus. The authors were able to validate the functionality of these signaling systems, characterize the activity of these LCPs as intercellular signals that are secreted and get reinternalized into the cytosol, and exhibit that these circuitries regulate diverse bacterial phenotypes and behaviors. Overall, this is an excellent contribution to the fields of quorum sensing and bacterial communication with far reaching implications to how signaling molecules are viewed and evaluated. I believe that this work is novel, very significant, and should be of interest to the broader scientific community. Furthermore, I found this work to be cutting-edge and meticulously executed, and therefore recommend it to be published in Nat. Commun. after the authors address a few minor issues:

- Generally, although the paper is well-written, it has some language issues throughout. I recommend that the authors carefully review the paper and correct these issues, or utilize a professional service to correct the text.

RESPONSE: We have proofed and edited the manuscript carefully to eliminate language issues and typos.

- Additionally, the text contains a few typos that need to be corrected. For example, on page 10, line 220, "...acts an intercellular..." should be "... acts as intercellular...".

RESPONSE: Corrected.

- Also, in the Figure legend of Figure S3 in the SI, "...whereas varying concentrations LCPss were used." should be "...whereas varying concentrations of FITC-LCPss were used."

RESPONSE: Corrected.

Reviewer #1 (Remarks to the Author):

This work continues to be novel and significant and the inclusion of an additional peptide/pheremone pair as well as statistical analysis improves the manuscript. Unfortunately, not being able to include a more distantly related pair does slightly decrease the significance of the findings in terms of how different it is from previously published work.

Reviewer #2 (Remarks to the Author):

In my opinion, the authors have responded to my questions and concerns, changing the manuscript accordingly. Particular, the analysis in a much more distant system gives much more robustness to the authors' conclusions. Therefore, the quality of the manuscript has substantially been improved and is now appropriate for publication.

Finally, I will add two minor comments on the answer to my questions about DNA binding which has been answered in a more superficial way by the authors.

- The authors compare the potential binding site of RopBss and RopB-GAS, but not information (or at least I do not find it in the manuscript) about RopB-GAS (from which strain is coming, which signal recognize, homology between both receptor and signals). This information could provide explanation for the differences in the binding sites of both receptors.

- Regarding this point, it has recently been published that RopB-GAS might act as a repressor and not as a transcriptional activator (DOI: 10.26508/lsa.202201809), as the authors show for RopBss. Could this fact explain the differences in their binding sites? Perhaps this point would be interesting to discuss in the manuscript.

RESPONSE TO REVIEWER'S COMMENTS:

Reviewer #1 (Remarks to the Author):

This work continues to be novel and significant and the inclusion of an additional peptide/pheromone pair as well as statistical analysis improves the manuscript. Unfortunately, not being able to include a more distantly related pair does slightly decrease the significance of the findings in terms of how different it is from previously published work.

RESPONSE: We acknowledge the reviewer's comments about the significance of including a more distantly related peptide/pheromone pair with peptide sequences distinct from SIP-like LCPs. However, as noted in the previous response, except for the characterized LCPs, nearly all the identified candidate LCPs are from strict anaerobes. Consequently, despite our efforts, we were unable to grow these bacteria in the available microaerophilic conditions, which prevented us from characterizing the distantly related LCP-receptor pairs.

Reviewer #2 (Remarks to the Author):

In my opinion, the authors have responded to my questions and concerns, changing the manuscript accordingly. Particular, the analysis in a much more distant system gives much more robustness to the authors' conclusions. Therefore, the quality of the manuscript has substantially been improved and is now appropriate for publication.

Finally, I will add two minor comments on the answer to my questions about DNA binding which has been answered in a more superficial way by the authors.

- The authors compare the potential binding site of RopBss and RopB-GAS, but not information (or at least I do not find it in the manuscript) about RopB-GAS (from which strain is coming, which signal recognize, homology between both receptor and signals). This information could provide explanation for the differences in the binding sites of both receptors.

RESPONSE: It is well documented that individual bacterial transcription factors within a large superfamily (such as TetR, MarR, DtxR etc.) share significant structural similarity and high sequence similarity within their DNA binding motifs. However, individual regulators within a family typically recognize entirely different DNA operator sequences (varying length and nucleotide composition, and location in the promoter) with unique binding modes (monomer to oligomers) and mechanisms of transcription regulation (recruiting different subunits of RNA polymerase to blocking repressors). Thus, comparisons of homology between receptors within a family are unlikely to yield explanation for the differences in the DNA binding specificity of each receptor.

Our and other published findings show that the amino acid sequence of RopB and its binding site in target (*SIP/speB*) promoters are conserved among nearly all the characterized GAS strains. We provided the sequence information for the signals (LCPs) (Fig. 4) and sequence similarity information between RopB sequences from GAS and *S. salivarius* in supplementary figure 2. However, as we noted above, such information will

not provide the desired explanation for the differences in the binding sites of both receptors. Consequently, we provided analyses and explanation in the previous version that is supported by the data and meaningful.

- Regarding this point, it has recently been published that RopB-GAS might act as a repressor and not as a transcriptional activator (DOI: 10.26508/lsa.202201809), as the authors show for RopBss. Could this fact explain the differences in their binding sites? Perhaps this point would be interesting to discuss in the manuscript.

RESPONSE: We are aware of this study. However, the indicated study failed to provide any direct evidence for apo RopB mediated promoter binding or identified DNA binding sites specific for RopB repression. Since the mechanistic or biochemical details for RopB-mediated repression was not provided in the indicated study, we did not discuss those findings in the manuscript.

Contrary to this, our detailed published findings show that RopB binds to specific DNA/promoter sequences with high affinity only in the presence of SIP and functions as an activator, not as a repressor. Thus, the comment that “RopB-GAS might act as a repressor and not as a transcriptional activator” is not correct and will not explain the differences in the binding sites of LCP receptors.